# Development of a Method to Determine the Fractional Deposition Efficiency of Full-Scale HVAC and HEPA Filter Cassettes for Nanoparticles ≥3.5 nm

**Ana Maria Todea [1,*], Frank Schmidt [2], Tobias Schuldt [2]**  **and Christof Asbach [1]**

[1]    Air Quality & Filtration Unit, Institut für Energie-und Umwelttechnik e. V. (IUTA),
      47229 Duisburg, Germany; asbach@iuta.de
[2]    Nanoparticle Process Technology, Faculty of Engineering, University of Duisburg-Essen,
      47057 Duisburg, Germany; frank.schmidt@uni-due.de (F.S.); tobias.schuldt@uni-due.de (T.S.)
*    Correspondence: todea@iuta.de

**Abstract:** Novel methods have been developed to measure the fractional deposition efficiency for nanoparticles of full-scale HVAC and HEPA filter cassettes down to a particle size of 3.5 nm. The methods use a flame spray nanoparticle generator to produce NaCl test aerosols with narrow size distributions and very high concentrations. While the efficiency curves of lower efficiency filters of classes F7 and E10 were still able to be determined by measuring the size distributions of the polydisperse test aerosols upstream and downstream of the filter, two new testing procedures were developed for high efficiency filters of class H13. One considers the narrow size distributions of the test aerosols as quasi-monodisperse and follows a similar approach like EN 1822 for flat sheet media. The second one evaluates mobility classified fractions of the quasi-monodisperse test aerosols. A dedicated multiple charge correction scheme was developed to account for the effect of multiply charged particles. While the latter procedure allows to extend the particle size range, the prior significantly reduces the measurement time. All tests delivered meaningful results, which were very comparable with the results from flat sheet media tests.

**Keywords:** nanoparticle; filtration; filter; HVAC; HEPA

## 1. Introduction

Airborne nanoparticles can cause adverse effects on human health [1–3] or affect the product quality [4]. By definition, nanoparticles are smaller than 100 nm [5]. They can stem from a variety of natural and anthropogenic sources. Natural sources include volcanoes [6], forest fires [7], and gas to particle conversion [8,9]. Anthropogenic nanoparticles can be differentiated into particles deliberately produced for certain applications like scratch-resistant coatings [10], lotus-leaf like self-cleaning surfaces [11] or energy applications [12] and unintentionally produced by-products, e.g., from traffic [13,14], industry emissions [15,16], candle and incense burning [17]. Unintentionally produced nanoparticles are often also termed ultrafine particles (UFP). Recently, due to the Covid-19 pandemic, the filtration of airborne viruses in HVAC (heating, ventilation and air conditioning) filters has raised increased attention. Many viruses are also in the nanoparticle size range, while the SARS-CoV 2 virus is on the edge with sizes between 60 nm and 140 nm [18].

HVAC systems are often used to supply filtered, clean air to indoor environments like residential homes, workplaces or clean production facilities. In the EU, the filters used in HVAC systems have previously been classified according to two different standards. The lower efficiency filters were classified according to the standard EN 779:2012 [19]. Classes G1 to G4, as well as M5 and M6, are typically used

in HVAC systems for residential buildings or workplaces as pre-filters for the final filters of classes F7 to F9. If a better air quality is required downstream, then filter classes E10 to E12 (EPA), H13 to H14 (HEPA), or U15 to U17 (ULPA) can be used which are classified according to EN 1822-1 in Europe [20]. Meanwhile EN 779 has been replaced by ISO 16890 [21] and EN 1822-2 to EN1822-5 by ISO 29463-2 to ISO 29463-5 [22]. The measurement procedures used in the new standards are mostly very similar to the ones in the old standards, but data evaluation and the filter classification schemes have partially changed. Since the experiments described here were carried out when the new standards were not yet in place, the nomenclature used in this paper is based on the old EN 779 standard.

In general, a higher filtration efficiency causes a higher pressure drop and thus besides higher investment also increased operational costs. ULPA filters are therefore mainly used to supply high class cleanrooms, e.g., for the manufacture of semiconductor chips or pharmaceuticals [23], where even single particles could do harm to the production facility or the products. Operation theaters in hospitals often use HEPA filters, which sometimes can also be found in residential HVAC systems or clean microenvironments. HEPA and ULPA filters are often protected by F7 to F9 pre-filters. Table 1 provides an overview of the requirements for different filters according to EN 779 and EN 1822-1. Typically, these filters have a cross sectional area of 592 mm × 592 mm and are operated at nominal flow rates between 1700 m$^3$/h and 4250 m$^3$/h.

**Table 1.** Classification into different filter classes according to EN 779 and EN 1822-1.

| Filter Class | F7 | F8 | F9 | E10 | E11 | E12 | H13 | H14 | U15 | U16 | U17 |
|---|---|---|---|---|---|---|---|---|---|---|---|
| Standard | | EN779 | | | | | | EN1822 | | | |
| min. Efficiency (%) | 35 [1] | 55 [1] | 70 [1] | 85 [3] | 95 [3] | 99.5 [3] | 99.95 [3] | 99.995 [3] | 99.9995 [3] | 99.99995 [3] | 99.999995 [3] |
| avg. Efficiency (%) | 40–60 [2] | 90–95 [2] | ≥95 [2] | n/a | n/a | n/a | n/a | n/a | n/a | n/a | n/a |

[1] lowest efficiency for 0.4 µm particles measured with a new filter, discharged filter and during filter aging. [2] average efficiency during filter aging with test dust until pressure drop reaches 450 Pa; for 0.4 µm particles. [3] minimum efficiency at the most penetrating particle size (MPPS).

The particle collection efficiency of a filter is always a function of the particle size. In purely mechanical filters, three main particle collection mechanisms prevail [24]. (1) Large and heavy particles are captured by impaction, i.e., they cannot follow the flow streamlines around a filter fiber and collide with the fiber because of their high inertia. The collection efficiency by impaction therefore increases with increasing particle size. (2) Very small particles in the nanometer size range are affected by Brownian motion and therefore move in a random zigzag motion, which makes them divert from the flow streamlines and collide with the fiber surface. This effect is also known as particle diffusion and increases with decreasing particle size. (3) A particle with a diameter $d_p$ that follows a flow streamline whose minimal distance to the fiber surface is $\leq 0.5 \cdot d_p$ will touch the fiber surface and be collected. This mechanism is known as interception and increases with increasing particle size. It is dominating in the intermediate size range, where diffusion is no longer and impaction not yet very efficient. The combination of these three mechanisms leads to the typical V-shape of the fractional filtration efficiency as a function of particle diameter as depicted in Figure 1. The particle size, for which the lowest collection efficiency occurs, is known as most penetrating particle size (MPPS). It should be noted that the graph shown is only a cartoon. The exact efficiencies and locations of the MPPS vary with filter media and geometry of the filter cassette. Typical HVAC filters have an MPPS around 200–300 nm, whereas the MPPS of high efficiency (HEPA and ULPA) filters is smaller, according to ISO 29463 mostly in the range from 120 nm to 250 nm.

The three aforementioned particle collection mechanisms may be augmented by electrostatic effects, e.g., in electret filters with charged fibers [25–28].

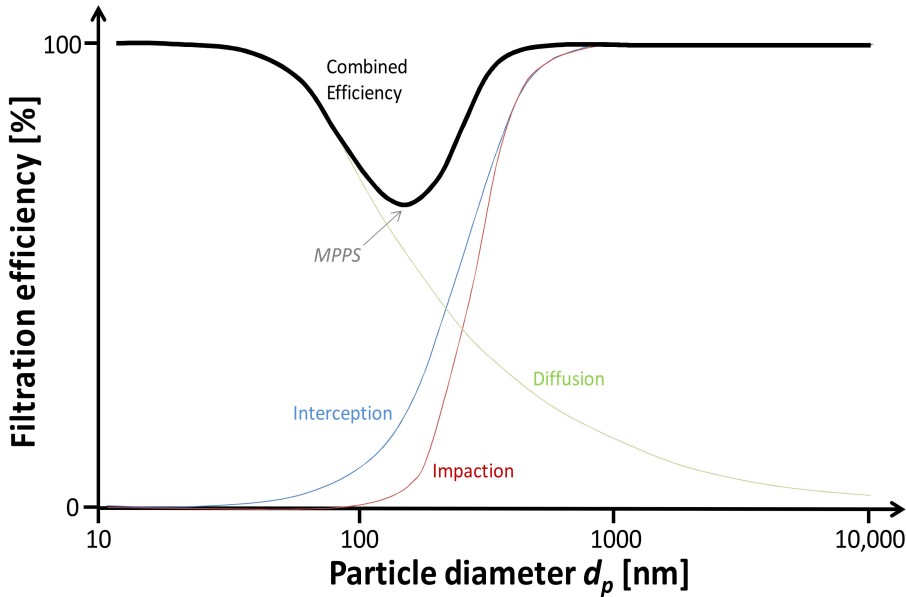

**Figure 1.** Exemplary fractional filtration efficiency of filters (adapted from [29]).

As can be seen from Figure 1, the filtration efficiency for small nanoparticles is very high, but decreases with increasing particle size until it reaches a minimum, before interception and impaction become dominant. Fractional filtration efficiency curves have been experimentally determined down to the nanometer size range for flat sheet media samples [30–34], wire screens [35–38], and respirators [39–41]. Recently, a method has been developed to determine the fractional filtration efficiency of flat sheet media in a particle size range from 3 nm to 500 nm and successfully undergone an inter-laboratory validation [42]. Experimental studies were also used to verify numerical data [43–45]. Overviews of nanoparticle filtration are given in [46] and [47].

According to EN 779 [19], the classes of full-scale HVAC filter cassettes (≤F9) are only rated based on the fractional filtration efficiency at a particle size of 0.4 µm. It is determined for a polydisperse DEHS test aerosol using an optical aerosol spectrometer. As of now, an established method to determine the fractional filtration efficiency of full-scale HVAC filters down to the nanometer size range is lacking. Published data on the filtration efficiency of HVAC filter cassettes for nanoparticles are therefore scarce. Hecker and Hofacre [48] investigated for the US Environmental Protection Agency (EPA), how the filtration efficiency of HVAC filters with different MERV classes (according to ASHRAE 52.2) changes with filter aging for a wide particle size range from 30 nm to 10 µm. In their study they used a potassium chloride (KCl) aerosol, but do not specify how it was generated. Their data were later used by Azimi et al. [49] to estimate the filtration efficiency of HVAC filters for ambient fine and ultrafine particles. Stephens and Siegel [50] determined the fractional filtration efficiency for particle sizes ranging from 7 nm to 100 nm in situ in HVAC systems with MERV4 to MERV16 filters, using ambient aerosol. The maximum efficiency found in their study was approximately 80%. According to EN 1822 only the measurement of the deposition efficiency in MPPS, as well as a leak test, is carried out for full-scale filters.

Neither EN 779 nor EN 1822 foresees the measurement of the fractional deposition efficiency curve of full-scale filters for nanoparticles. The main challenge in the determination of the fractional filtration efficiency for nanoparticles, especially in case of high efficiency filters is that the high filtration efficiency (see Figure 1) necessitates the provision of a sufficient upstream concentration of these particles at the nominal flow rates of the filters in order to still be able to measure the size distribution or at least the number concentration of the particles downstream. Common aerosol generators like atomizers are therefore not suitable, because the particle concentration is too low after dilution with the operational filter flow rate. In general, the test aerosols may be either polydisperse or (quasi-)

monodisperse. In case of a polydisperse aerosol, the particle size distribution can be measured upstream and downstream of the filter, e.g., with a scanning mobility particle sizer (SMPS [51]) to determine the fractional filtration efficiency in the nanometer size range. An SMPS, however, requires a minimum particle concentration (depending on the SMPS components, settings and particle sizes, but as a rule of thumb around 1000 $1/cm^3$) and therefore this method fails if the filtration efficiency is very high. In this case, the fractional filtration efficiency can only be determined by using monodisperse particles and counting the particles downstream, e.g., with a condensation particle counter (CPC). Modern CPCs have very low false count rates so that concentrations can still reliably be determined, even if only a very low number of particles are counted downstream. By using this procedure, each measurement yields a single data point of the fractional filtration efficiency curve for the size of the monodisperse particles used. The particle sizes of the monodisperse test aerosols therefore need to be adjustable in order to construct the fractional filtration efficiency curve. This method is comparable to the one described in EN 1822-3 [52] for high efficiency flat sheet media. However, currently no aerosol generation system exists that would be capable of producing a sufficiently high concentration of monodisperse (nano-) particles to test full-scale HVAC filter cassettes.

## 2. Experiments

### 2.1. Experimental Setup

In order to produce the high concentrations needed to determine the fractional filtration efficiency of HVAC and HEPA filters for nanoparticles, a flame spray nanoparticle generator (Model FG2, MoTec Konzepte, Bochum, Germany [53]) was used in connection to a filter test rig according to EN 779/ISO 16890. The test rig can be air conditioned and is operated in a loop at a slightly negative pressure. In the nanoparticle generator, an aqueous sodium chloride (NaCl) solution is continuously sprayed via a binary nozzle into a hydrogen-oxygen flame (10 L/min $H_2$ and 5 L/min $O_2$), where the solution completely evaporates [54]. While the generator can also be used to produce other nanoparticles, NaCl was chosen here, because it is readily available and harmless, in case it gets emitted into the laboratory atmosphere. The feed rate of the solution is controlled by a syringe pump. Upon cooling, the NaCl nucleates to form new and very small particles. It is known that nucleation generates very high concentration of particles [55,56], which can easily exceed $10^{10}$ $1/cm^3$. Subsequently, the remaining NaCl vapor condenses onto the newly formed NaCl particles, which makes the particles grow. Vapor condensation onto seed particles produces (nearly) monodisperse particles [57,58]. By quenching the nucleation and condensation process early on, the particles remain very small. In the setup used here, the quenching is realized by feeding the freshly produced aerosol directly from the generator into the filter test rig, where it mixes with the total air flow rate (see Figure 2) used for filter testing. The flow rate inside the test rig should equal the nominal flow rate of the test filter. Typical nominal flow rates of full-scale HVAC/HEPA filters are 1700 $m^3/h$, 3400 $m^3/h$ and 4250 $m^3/h$, respectively. The centerpiece of the test rig (model ALF 114, Topas GmbH, Dresden, Germany), which holds the test filter has a square cross sectional area of 610 mm × 610 mm. The remaining (homemade) parts of the filter test rig consist mainly of circular tubes with a diameter of 400 mm. The freshly produced aerosol enters the test rig through a 110-mm diameter opening in the side of the test rig, to which an extension tube can be attached. Near the transition from the circular to the rectangular section, a baffle plate is installed to spatially homogenize the particle concentration.

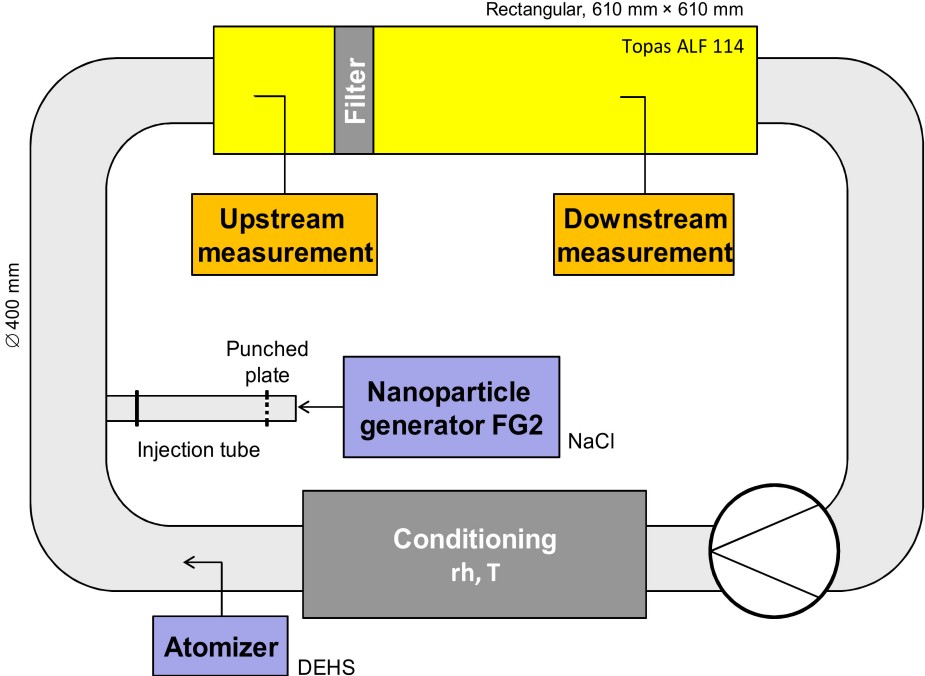

**Figure 2.** Test rig according to EN779/ISO16890 with attached nanoparticle generator FG2 or atomizer.

The size of the particles can be controlled by adjusting the NaCl concentration in the solution and the feed rate of the solution. The lowest NaCl concentration that still delivers a constant output is 0.5 g/L. Concentrations higher than 80 g/L lead to the rapid clogging of the binary nozzle and should thus be avoided. In addition, the particle size can be controlled via the residence time of the freshly produced aerosol in the injection tube before it mixes with the high flow rate of the test rig. The residence time can be varied by adjusting the length of the injection tube. However, as listed in Table 2, the flow rates in the injection tube are rather high, so that a simple extension of the tube changes the particle sizes only negligibly. To further increase the residence time, an additional punched plate can be added to the injection tube, thereby increasing the pressure drop and decreasing the flow rate drawn through the tube by the negative pressure of the test rig. Two different punched plates were used, one with 25 holes and the other with 48 holes with diameters of 3 mm each. Flow velocities in the injection tube were measured three times with a hot wire anemometer (model 425, Testo AG, Titisee-Neustadt, Germany). Table 2 provides an overview of the different configurations of the injection tube and the resulting mean flow rates and residence times in the injection tube and their standard deviations.

**Table 2.** Aerosol injection flow rate and residence time of the aerosol in the injection tube for different configurations with or without the punched plate.

| Flow Rate Test Rig [m³/h] | Injection Tube Length [mm] | Punched Plate | Injection Flow Rate [m³/h] | Residence Time [s] |
|---|---|---|---|---|
| **3400** | 338 | none | 561.0 ± 7.04 | 0.021 ± 0.0003 |
| | 840 | none | 632.6 ±4.46 | 0.047 ± 0.0003 |
| | 1400 | 25 holes | 24.7 ± 0.20 | 2.010 ± 0.0167 |
| | 1400 | 48 holes | 65.4 ± 0.20 | 0.759 ± 0.0024 |
| **4250** | 338 | none | 631.6 ± 14.64 | 0.019 ± 0.0004 |
| | 840 | none | 746.1 ± 0.74 | 0.040 ± 0.0000 |
| | 1400 | 25 holes | 29.9 ± 0.41 | 1.660 ± 0.0229 |
| | 1400 | 48 holes | 80.3 ± 0.54 | 0.619 ± 0.0042 |

In parallel, the fractional filtration efficiency of media samples was determined with the setup shown in Figure 3 by adjusting the same media face velocity as in the filter cassette. Media samples from the same batch as used to produce the filter cassettes were provided by the filter manufacturers to assure the highest possible degree of comparability among the tested materials. The setup mainly consists of a modified HEPA filter test stand (TSI, model 8130) according to EN 1822. The filter tester comprises an aerosol generation section (1) with three atomizers that may be used to produce and mix three different size distributions at a time. Here, only a single atomizer was used to produce Di-Ethyl-Hexyl-Sebacate ($C_{26}H_{50}O_4$, DEHS) aerosols. The size distribution of the DEHS aerosol was modified by diluting the DEHS liquid with different amounts of isopropyl alcohol (IPA) prior to atomization. The aerosol generation system has been modified so that the polydisperse aerosol can alternatively be produced by a spark generator (GFG digital 3000, Palas GmbH, Karlsruhe, Germany), equipped with homemade silver electrodes. In this configuration, the spark generator produces very small particles with sizes down to <10 nm at high concentrations of >$10^7$ 1/cm$^3$. The aerosol generation section is followed by an $^{85}$Kr neutralizer (2) to bring the particles to bipolar charge equilibrium before they are classified in a differential mobility analyzer (DMA) (3). Downstream of the DMA, the particles are neutralized again in a soft x-ray neutralizer (HTC 4530, Seoul, Korea), before they reach the filter (4). The filter holder has an open filter area of 100 cm$^2$, as requested in EN 1822-3. Particle concentrations of the monodisperse aerosol are measured upstream and downstream of the filter with two separate condensation particle counters (CPCs) (TSI model 3022 and 3025A), that had undergone a thorough study to compare their performance [59].

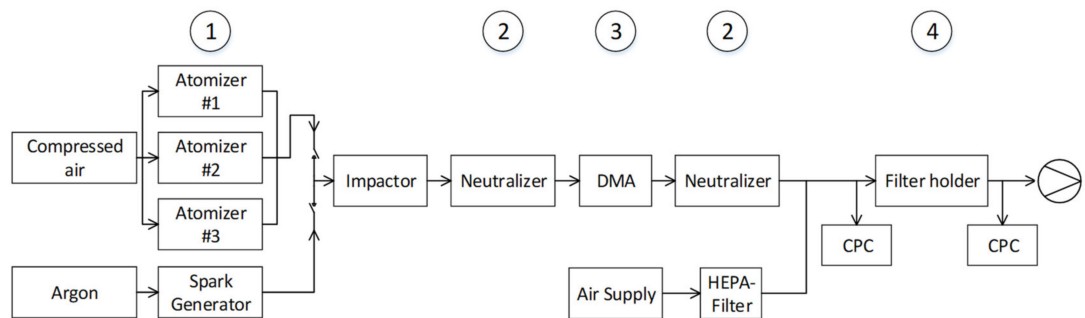

**Figure 3.** Experimental setup for testing filter media samples; (1) polydisperse aerosol generation, (2) neutralization, (3) size/mobility classification, (4) filter holder with upstream and downstream CPC.

*2.2. Size Distributions of the Test Aerosols*

The size distributions and total number concentrations inside the test rig upstream of the filter elements have been measured for the two most common flow rates of HVAC filters, i.e., 3400 m$^3$/h and 4250 m$^3$/h, respectively, for different configurations of the aerosol generation and injection (see Table 2). Measurements were carried out with a scanning mobility particle sizer (SMPS, TSI model 3936) with either nano DMA (TSI model 3085) or long DMA (TSI model 3081) and a butanol-based ultrafine condensation particle counter (UCPC, TSI model 3776) or a water-based CPC (TSI model 3787), depending on the availability of the CPCs. The produced size distributions and concentrations showed a high temporal stability with only short interruptions, when the syringe with the NaCl solution was replaced. Exemplary normalized size distributions are depicted in Figure 4 for aerosol injection without (top) and with (bottom) the punched plates. The figure shows that changing the feed rate of the sodium chloride solution changes only slightly the particle size (black curves in top graphs), whereas a change in the sodium chloride concentration has the main effect on the generated particle size (red curves in top graphs). The smallest particles that are generated without the punched plates can be as small as approximately 4.4 nm (modal diameter) at a flow rate of 3400 m$^3$/h and 3.5 nm at 4250 m$^3$/h, respectively. Test aerosols with such small particles, however, are less stable than the ones with larger particle as indicated by the larger error bars. Adding a longer injection tube led to

only very small differences in the size distributions, as could be expected from the still very short residence time in the injection tube (see Table 2). When a punched plate was installed, the flow rate through the injection tube was strongly reduced and consequently the residence time increased. The maximum modal diameters that could be produced this way were 79.1 nm at 3400 m$^3$/h and 68.5 nm at 4250 m$^3$/h, respectively, with a NaCl concentration in the solution of 80 g/L and a feed rate of 90 mL/h (see bottom graphs in Figure 4). The data of the lognormal size distributions, generated as test aerosols are summarized in Table S1 in the Supplementary Materials. The geometric standard deviations are small and mostly below 1.5. The test aerosols are therefore not strictly monodisperse, but can be considered to be quasi-monodisperse according to VDI standard 3491-1 [60]. All number concentrations were well above 10$^6$ 1/cm$^3$. At this concentration, it is likely, that the particles have agglomerated while they traveled from the generator to the filter element. However, characterization of the particles with electrical mobility analysis (e.g., with a scanning mobility particle sizer [51]) yields their mobility equivalent diameter of the aspherical particles, which also describes particle diffusion, i.e., the dominating filtration mechanism in this size range. Nevertheless, particles may further coagulate as they travel from the plane of the aerosol sampling port to the filter element, which is approximately 600 mm downstream. Based on the data provided in Table S1 the change of the particle number concentration and mean particle size were estimated to be mostly below 10%. In order to compensate for coagulation changes and to assure that the measured size distributions are identical with those in the filter element, the particle residence time in the sampling train from the sampling point inside the test rig to the measurement instrument was adjusted to equal the time particles need to travel from the sampling point to the filter element. Consequently, the test aerosol would encounter the same changes in the sampling train as inside the filter test rig.

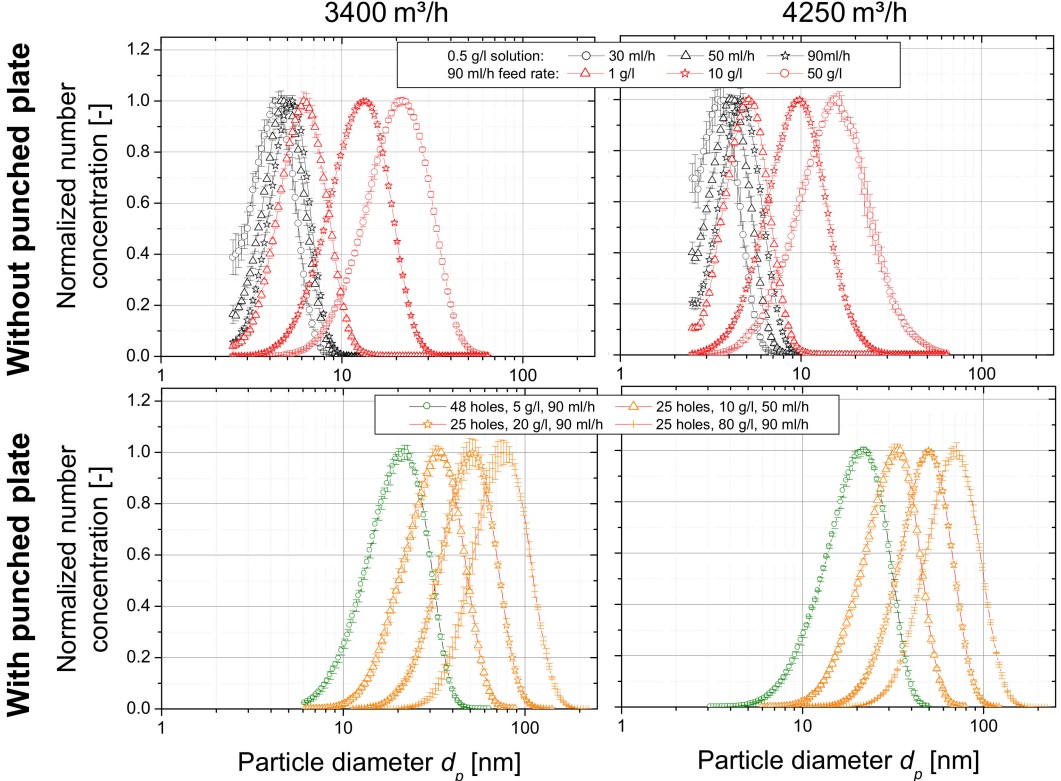

**Figure 4.** Exemplary normalized number size distributions, measured inside the test rig, upstream of the filter element at two different air flow rates: 3400 m$^3$/h (**left**) and 4250 m$^3$/h (**right**) without (**top**) and with punched plate (**bottom**); top: 0.5 g/L aqueous NaCl solution at three different feed rates (black), three different NaCl solutions at feed rate of 90 mL/h (red); bottom: injection with punched plate with 48 holes (green) and 25 holes (orange).

The spatial homogeneity of the particle concentration inside the test rig has been measured up to nine positions at the location where normally the filter element is installed. For flow rates of 3400 m$^3$/h and 4250 m$^3$/h the max. deviations from the concentration in the center were found to be +7.2% and +7.9%, respectively. The deviation was a bit higher (+11.8%) in case of the flow rate of 1700 m$^3$/h. The data can be found in the Supplementary Information and in Figure S1 of the Supplementary Materials.

Typically, the most penetrating particle size (MPPS) of HVAC and HEPA filters is in the range between 100 nm and 400 nm and cannot be covered with the nanoparticle generator used in this study. Instead, an atomizer (AGF 2.0B, Palas GmbH, Karlsruhe, Germany) was connected to the test rig to produce a polydisperse DEHS aerosol and the particle size distribution upstream and downstream of the filter, as e.g., prescribed in EN 779 [19] and ISO 16890 [21], was measured.

### 2.3. Test Procedures

Two procedures to determine the fractional deposition efficiency of a filter are most commonly used. Either a polydisperse aerosol is produced, its size distribution measured upstream and downstream of the filter and the fractional deposition efficiency determined from size differentiated ratio of the upstream and downstream particle concentration. This method fails, if the filtration efficiency is very high and/or the upstream concentration low, because in these cases the downstream particle concentration is too low for most available aerosol measurement equipment to determine a size distribution in the nanoparticle size range. Instead, monodisperse particles can be used. In such cases it is sufficient to determine only the number concentration upstream and downstream, which can still be accurately done at very low concentrations. The main downside of using monodisperse particles is that it is a lot more laborious, because the efficiency has to be determined separately for each particle size of interest. Because of the high filtration efficiency, the fractional deposition efficiency for nanoparticles can often only be determined by using monodisperse particles. For filter media testing, monodisperse particles are usually provided by classifying particles of the wanted size from a polydisperse aerosol by employing a DMA [61]. A main challenge, when using a DMA to provide monodisperse particles is the fact that a DMA classifies particles based on their electrical mobility which is a function of both the particle size and the particle charge. Therefore, different combinations of particle size and charge yield the same electrical mobility. DMA-classified particles are therefore primarily monomobile and not monodisperse. Commonly, the wanted monodisperse particle size is the size of particles carrying a single elementary charge. The fraction of multiply charged particles in the classified monomobile aerosol depends on the combination of the size distribution of the polydisperse aerosol, the wanted monodisperse particle size, as well as the charge distribution of the polydisperse aerosol. The latter is well defined and in bipolar equilibrium, if an appropriate neutralizer is used [62–64]. The charge distribution of the particles is getting broader, the larger the particles and thus the amount of multiply charged particles in the classified monomobile aerosol is generally increasing with increasing particle size [65] and thus the distribution can no longer be deemed monodisperse. This effect can drastically bias filtration efficiency measurements [66]. In part, this can be overcome by choosing an adequate polydisperse size distribution with a modal diameter smaller than the wanted monodisperse particle size [67]. While the method of providing monodisperse particles with a DMA is common for media tests [42,47] at relatively low flow rates and required in tests according to EN 1822-3 [52], it is not suitable for determining the fractional filtration efficiency of full-scale HVAC or HEPA filters, because the concentrations of monomobile particles downstream of a DMA are rather low. Additionally, the highest flow rates of a DMA are at most between 6 m$^3$/h (100 L/min [68]) and 30 m$^3$/h (500 L/min [69]). After dilution with the operational flow rate of a full-scale filter inside the filter test rig, the concentrations would be too low for the measurements. Instead, we here followed two different approaches. On the one hand, the size distributions produced with the setup described above are rather narrow and can be considered as quasi-monodisperse (see Table S1 of the Supplementary Materials) [60]. If the monodispersity of the

test aerosol is considered sufficient, the number concentration of the test aerosol of known particle size can be measured upstream and downstream of the filter and the fractional efficiency can be calculated. The data can then be plotted using the count median diameter as particle size and the geometric standard deviation of the quasi-monodisperse test aerosol indicated by error bars.

Alternatively, if the geometric standard deviation of the test aerosol is considered to be too large and a filter evaluation with more monodisperse particles is desired, we developed a new method, which is based on the multiple charge correction of the differential mobility particle sizer by Hoppel [70]. The filter is challenged with a test aerosol produced with the nanoparticle generator. Low sample flow rates are withdrawn from the filter test rig upstream and downstream of the filter and the number concentration of only a monomobile fraction is determined. To do so, the sampled aerosol is neutralized and classified with a DMA at a fixed voltage. With this procedure, the challenge of multiply charged particles occurs in the same way as it does during filter media tests as described above. The DMA voltage is set such that the size of the singly charged particles matches the size of the wanted monodisperse particles. Neglecting particle losses, the concentration of these particles in the classified aerosol equals the concentration of particles of this size in the polydisperse aerosol multiplied with their probability to carry a single elementary charge. However, the classified aerosol may also contain fractions of larger doubly and triply charged particles. To determine their concentrations and use them for the multiple charge correction, the DMA is next set to select particles of the sizes of the doubly and triply charged particles. The probability of a particle of a given size to carry a certain number of elementary charges can be estimated by the Wiedensohler approximation [63] of the Fuchs charge distribution [62]. In the particle size range considered here, the probability of a particle to carry two or more elementary charges is always much lower than the probability to carry only a single elementary charge. Nevertheless, a well-defined starting point is needed for the multiple charge correction, which in the case of an SMPS or DMPS is provided by the use of an appropriate impactor [71]. There, the impactor assures that the largest size bin only contains singly charged particles, because larger multiply charged particles would remain in the impactor. Here, the use of an impactor with a cut-off well in the nanometer size range is excluded, because it would introduce too high a pressure drop. However, when classifying a particle size $d_{p1}$ much larger than the modal diameter from a very narrow, quasi-monodisperse aerosol, the concentration at the size $d_{p2}$ of the doubly charged particles with the same electrical mobility is so low that it can be considered to be zero and hence the concentration determined at $d_{p1}$ only contains singly charged particles and is thus monodisperse [67]. This value can hence be taken as the starting point for the multiple charge correction and the number concentration at several distinct (smaller) sizes can be determined. Since the size distributions produced with the nanoparticle generator are rather narrow (see Table S1), the subsequent use of several test aerosols is needed to construct the fractional filtration efficiency curve.

## 3. Results and Discussion

In order to test and validate the developed method, the fractional filtration efficiency has been determined for filters of classes F7 and E10 at their nominal flow rate of 4250 m$^3$/h. Two specimens of each filter type were tested. In addition, a H13 filter with a nominal flow rate of 3000 m$^3$/h was tested. Note that this flow rate is slightly lower than the ones for which the size distributions and their parameters are provided in Figure 4 and Table S1. All filters were factory new and used as they were without prior conditioning or discharging. All tests were conducted using the setup shown in Figure 2. NaCl particles were produced with the nanoparticle generator FG2 as explained above. In order to additionally receive information on the MPPS of the F7 and E10 filter, DEHS aerosol was produced by using an atomizer and the size distribution measured upstream and downstream of the filters. This method to determine the MPPS worked reasonably well up to the filter class E10, but failed for the H13 filter, because of too low downstream concentrations.

### 3.1. F7 Filters

For tests of full-scale filters of classes F7 and E10, the size distributions of the NaCl test aerosol could still be measured downstream. For each filter at least three raw gas/clean gas/raw gas measurements were performed for each test aerosol. However, due to the low concentrations at the upper and lower end of the distributions, the data could only be evaluated for the particle sizes around the modal diameter. It was empirically found that meaningful data could be derived at least from the size range $\frac{d_{mode}}{1.5} \leq d_p \leq 2 \cdot d_{mode}$. The fractional filtration efficiency curve is hence constructed using different NaCl and one DEHS size distributions. The same aerosol generator settings were used for all tests and are listed in Table 3.

**Table 3.** Settings and size distribution parameters of the test aerosols used for full-scale filter testing.

| Aerosol Substance | | NaCl-1 | NaCl-2 | NaCl-3 | NaCl-4 | NaCl-5 | NaCl-6 | NaCl-7 | DEHS |
|---|---|---|---|---|---|---|---|---|---|
| **3000 m³/h** | Solution [g/L] | 1 | 1 | 5 | 10 | 20 | 40 | 80 | n/a |
| | Solution feed rate [mL/h] | 40 | 99 | 90 | 50 | 90 | 90 | 90 | n/a |
| | Punched plate | 48 holes | 48 holes | 48 holes | 25 holes | 25 holes | 25 holes | 25 holes | n/a |
| | Mode [nm] | 7.9 | 11.5 | 19.4 | 29.4 | 49.2 | 60.4 | 74.3 | n/a |
| | GSD | 1.37 | 1.42 | 1.45 | 1.47 | 1.42 | 1.39 | 1.4 | n/a |
| | Number Conc. [1/cm³] | $1.2 \times 10^7$ | $8.7 \times 10^6$ | $6.2 \times 10^6$ | $1.7 \times 10^6$ | $1.2 \times 10^6$ | $1.4 \times 10^6$ | $1.3 \times 10^6$ | n/a |
| **4250 m³/h** | Solution [g/L] | 1 | 1 | 5 | 10 | 20 | n/a | n/a | pure |
| | Solution feed rate [mL/h] | 50 | 90 | 90 | 50 | 90 | n/a | n/a | n/a |
| | Punched plate | none | 48 holes | 48 holes | 25 holes | 25 holes | n/a | n/a | n/a |
| | Mode [nm] | 5.9 | 11.8 | 20.9 | 34.6 | 51.4 | n/a | n/a | 233 |
| | GSD | 1.34 | 1.42 | 1.47 | 1.42 | 1.4 | n/a | n/a | 1.81 |
| | Number Conc. [1/cm³] | $3.4 \times 10^7$ | $8.9 \times 10^6$ | $6.6 \times 10^6$ | $1.6 \times 10^6$ | $1.6 \times 10^6$ | n/a | n/a | $1.4 \times 10^4$ |

Figure 5a shows such results exemplarily for two identical F7 filter cassettes, measured at their nominal flow rate of 4250 m³/h. The filters were tested with five different NaCl test aerosols and one DEHS aerosol. The smallest particles that could be evaluated with test aerosol NaCl-1 were as small as 3 nm. As can be seen, the fractional filtration efficiency curve measured with the different NaCl test aerosols match well in the overlapping size range, but there is a small mismatch between the curves measured with NaCl and DEHS. A similar mismatch was consistently found for nearly all filters investigated. Possible reasons may be a different collection efficiency because of the different particle morphologies (cubic and agglomerated NaCl vs. liquid DEHS droplets) or a different reaction of the measurement techniques used to either the different particle materials or the different concentration levels. However, the number concentrations in the overlapping size range (test aerosols NaCl-4, NaCl-5 and DEHS, see Table 3) are well within the concentration range of the measurement equipment used in these measurements.

The graph in Figure 5a furthermore shows rather smooth curves for the filtration efficiency measured with NaCl aerosol, whereas the data measured with DEHS are more scattered. This is because of the much lower particle concentration of the DEHS aerosol compared with the NaCl aerosols (Table 3) which caused a larger statistical uncertainty in the size distribution measurements with the SMPS, especially downstream of the filter. This also shows that measuring the fractional filtration efficiency curve using an atomizer instead of the nanoparticle generator FG2 for particles smaller than the DEHS particles used here would not lead to meaningful results, because the downstream concentration would be reduced even further because of the higher filtration efficiency.

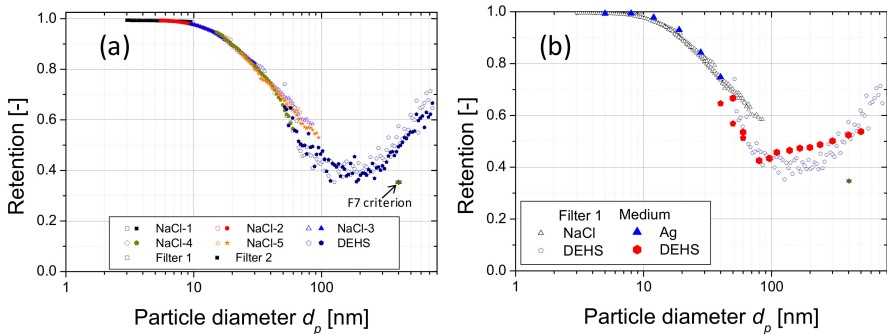

**Figure 5.** Exemplary results for (**a**) two identical full-scale F7 filter cassettes, flow rate 4250 m³/h, tested with different NaCl and DEHS aerosols (see Table 3); open symbols: filter 1, closed symbols: filter 2; (**b**) comparison of results from filter cassette 1 with data obtained from media sample.

The fractional efficiency curves of the two filters fall on top of each other, showing the validity of the method and the good reproducibility of the results. Figure 5b compares the results obtained with the filter cassette 1 with the data measured for the identical filter medium using the setup shown in Figure 3, tested with the same face velocity as for the full-scale filter. Again the results from both tests compare very well. Minor, yet noticeable differences can only be seen for DEHS particles larger than approximately 100 nm. These differences are likely caused by multiply charged, larger particles in the monomobile particle fraction downstream of the DMA, which bias the filtration efficiency [66]. The criterion for a filter to be rated as class F7 is that the efficiency at 400 nm is at least 35%. Figure 5 shows that the tested filters fulfill this requirement.

## 3.2. E10 Filters

Figure 6a shows the same type of data for two identical E10 filters, tested at their nominal flow rate of 4250 m³/h. Again, the results from the different NaCl size distributions agree well in the overlapping size ranges and the data from the two identical filters agree very well with each other. As in the case of the F7 filters, there is a small mismatch between the data obtained with NaCl and DEHS test aerosols. The data obtained with the full-scale filter cassette are also in very good agreement with the data obtained from media samples, except for the same deviation observed previously for DEHS particles larger than approximately 100 nm. Astonishingly, although the measured efficiencies are higher than for the F7 filters (Figure 5), both full-scale filters, as well as the filter media, did not fulfill the requirements for an E10 filter. According to EN1822, an E10 filter has to have a minimum efficiency of at least 85%, whereas here, the efficiency in the MPPS was only around 65% (filter cassette) and 70% (medium), respectively. The reason for this is unclear, but it is likely that the filter was wrongly labeled.

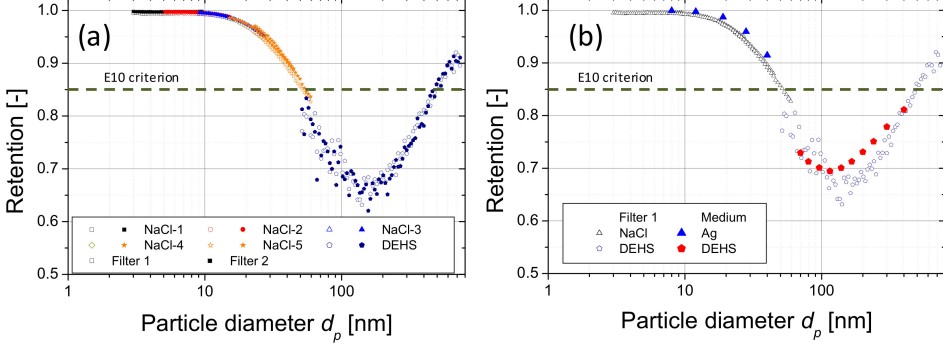

**Figure 6.** Exemplary results for (**a**) two identical E10 filter cassettes, flow rate 4250 m³/h, tested with different NaCl and DEHS aerosols (see Table 3); open symbols: filter 1, closed symbols: filter 2; (**b**) comparison of results from filter cassette 1 with data obtained from media sample.

### 3.3. H13 Filters

During tests with the H13 filter, downstream concentrations were too low to still be able to measure the size distributions with an SMPS. Instead the two different methods, as described above were used, i.e., on the one hand the NaCl size distributions from the flame spray generator were considered as quasi-monodisperse and their count median diameter used to represent the particle size. On the other hand, the filters were challenged with the aerosol and only the concentrations were measured with distinct electrical mobilities upstream and downstream using a combination of a nano DMA or a long DMA and CPC. The tailored multiple charge correction was applied as described above in order to obtain the data for these particle sizes. The advantage of the latter principle is that it allows to cover a wider particle size range, but the downside is that the measured concentrations are lower because of the classification and consequently the measurement duration increases. The data are shown in Figure 7.

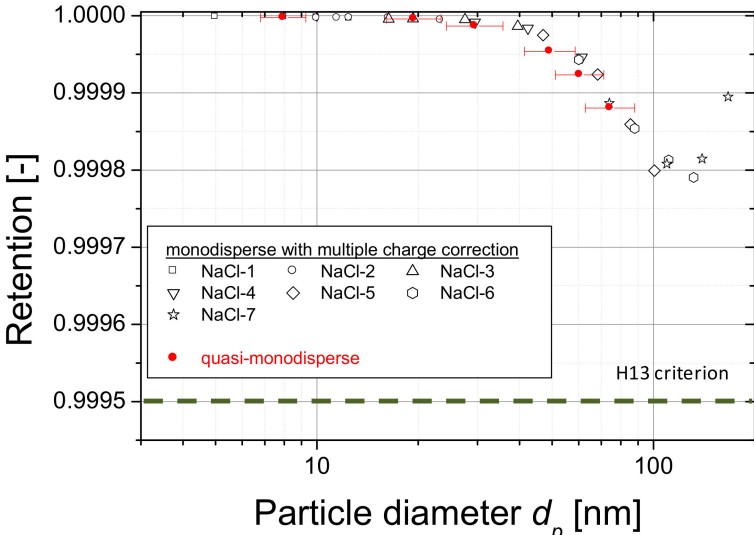

**Figure 7.** Results for the fractional filtration efficiency curve of an H13 filter cassette, determined with quasi-monodisperse NaCl aerosols (red) and for monodisperse aerosols (black), classified from quasi-monodisperse NaCl aerosols with modal diameters as given in the legend; error bars indicate the geometric standard deviation of the quasi-monodisperse test aerosols.

The largest particle size that could be evaluated was 167 nm, classified from a quasi-monodisperse aerosol with a modal diameter of 74.3 nm and a geometric standard deviation of 1.40. The figure first shows a good agreement between the fractional filtration efficiencies determined for the monodisperse particles for the overlapping particle sizes. The MPPS was found to be around 130 nm. The data, obtained with the quasi-monodisperse aerosol were overall in good agreement with the data obtained for monodisperse particles, however, since the largest modal diameter that could be produced with the current setup was 74.3 nm, the MPPS could not be determined with this method.

## 4. Conclusions

Methods to determine the fractional deposition efficiency of full-scale HVAC and HEPA filters with a cross sectional area of ~ 600 mm × 600 mm for nanoparticles with sizes down to 3.5 nm (depending on generator settings, aerosol conditioning and filter flow rate) have been developed. The methods use a flame spray aerosol generator to produce a NaCl test aerosol with adjustable median diameter. By quenching the particle formation and growth early on, the particles remain very small and their size distributions are narrow with geometric standard deviations below 1.5. The test aerosol can hence be considered to be quasi-monodisperse. Despite the strong dilution of the freshly produced

aerosol, the particle concentrations that reach the filter are still very high and between $2 \times 10^6$ 1/cm$^3$ and $>10^7$ 1/cm$^3$, thus allowing for the determination of even very high filtration efficiencies. It was consequently possible to measure the particle size distribution with an SMPS downstream of F7 and E10 filters and construct the fractional deposition efficiency curve by measuring a combination of multiple highly concentrated test aerosols. The efficiency curve could be extended up to 700 nm by using an additional polydisperse DEHS test aerosol. The results obtained with the full-scale HVAC filters agreed very well with those from the identical filter media tests.

This method failed for the test of a full-scale H13 filter, because the downstream particle concentrations were too low to measure size distributions. Instead, two different methods were developed. The first method considers the test aerosols produced by the flame spray generator to be quasi-monodisperse (justified according to VDI 3491-1) so that it is sufficient to measure the number concentration upstream and downstream of the filter. The resulting efficiency is presented as a data point of the fractional deposition curve for the geometric mean diameter (GMD) of the quasi-monodisperse test aerosol used and the geometric standard deviation represented as error bars. By the stepwise increase of the GMD of the test aerosol, the fractional deposition efficiency curve can be constructed up to the max. possible GMD of approximately 80 nm. A second method has been developed that can be applied if the monodispersity of the test aerosols is considered to be insufficient. In this method, the aerosol samples withdrawn upstream and downstream are first classified by a differential mobility analyzer (DMA) prior to the measurement of the particle number concentration. Consequently, the fractional deposition efficiency for multiple sizes can be determined with a single test aerosol. Therefore, with this method also the efficiency for particles larger than the GMD of the largest test aerosol can be determined. However, because of the lower concentrations, measurements require more time to deliver statistically significant results than measurements according to the first method. A dedicated multiple charge correction scheme has been developed to account for the effect multiply charged, larger particle downstream of the DMA. Both methods have been applied to a HEPA H13 filter and delivered very comparable results.

In summary, novel methods have been developed that allow for the determination of the full fractional deposition efficiency curves for nanoparticles of full-scale HVAC and HEPA filters down to 3.5 nm. The methods developed for high efficiency filters for the first time provide the opportunity to determine the MPPS of a full-scale HEPA filter cassette, which in the past has only been possible for filter media.

**Supplementary Materials:** The following are available online at http://www.mdpi.com/2073-4433/11/11/1191/s1, Figure S1: Determination of the spatial homogeneity of the particle concentration in the test rig; (a) matrix of nine spatially distributed measurement positions, view in flow direction; (b) deviations of the number concentrations measured in comparison to Pos. 5 at a flow rate of 1700 m3/h; (c) deviations of the number concentrations measured in comparison to Pos. 5 at a flow rate of 3400 m3/h; (d) deviations of the number concentrations measured in comparison to Pos. 5 at a flow rate of 4250 m3/h., Table S1: Parameters of the lognormal size distributions of the generated test aerosols for different settings.

**Author Contributions:** A.M.T. was responsible for the study, she carried out and evaluated the measurements on full-scale filters; A.M.T. and C.A. developed the test methods for full-scale filters; T.S. carried out and evaluated measurements on filter media; F.S. supervised filter media measurements; C.A. drafted the manuscript. All authors have read and agreed to the published version of the manuscript.

**Funding:** The research project was carried out in the framework of the industrial collective research programme (IGF no. 18314 N). It was supported by the Federal Ministry for Economic Affairs and Energy (BMWi) through the AiF (German Federation of Industrial Research Associations e.V.) based on a decision taken by the German Bundestag.

**Acknowledgments:** The authors would like to thank the participants of the project-accompanying committee for fruitful discussions and supply of the filter materials used in the project.

**Conflicts of Interest:** The authors declare no conflict of interest.

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
