# Peer review of "Development of a Method to Determine the Fractional Deposition Efficiency of Full-Scale HVAC and HEPA Filter Cassettes for Nanoparticles ?3.5 nm"

_atmosphere, doi:10.3390/atmos11111191_

Round 1

Reviewer 1 Report

Thank you for the interesting and nicely written manuscript. The authors present quite clear their aim and explain the methodology.

In the following, I give a few comments and suggestions:

1) Please correct the references to the figures and tables. In the present manuscript, it is only mentioned: "Error! Reference source not found."

2) Are the considerations restricted to fibrous filters? In the lines 62 to 75, the capturing mechanisms for fibrous filters are described. However, also membranes are used for HVAC and HEPA filters. Can the methods also used for those filters?

3) Has the measuring device also some pressure sensors? In line 138 it is stated that the test rig is operated under negative pressure. Is this checked before the measurements? Is the evolution of the pressure drop due to possible clogging of the filter device tracked? Is there an increase of the pressure drop due to those effects?

4) In the lines 59-60 it is stated, that the typical cross-section area is 592mmx592mm. In the experimental setup, the cross-section is slightly larger (line 154). It is only a small difference but is there a reason for that? In the supplement the stated size is again larger (610mm x 610mm). Can you please explain the differences?

5) In table 2, what is the resulting aerosol size for the different configurations?

6) For me it is not completely clear how the experimental setups in figure 2 and figure 3 are connected. Why the cross-section in the second setup is much smaller than the cross-section in the first one? Is only the second setup used to measure filtration efficiencies? If yes, why not the first one?

7) In line 214 it is mentioned that the distributions are normalized. In which way they are normalized? The figures indicate that the normalization is performed with the maximum of the curve. Why the normalization is not performed with respect to the total number of particles?

8) Figure 4: Are the variations in the measurements with the punched plate that small such that no error bars are shown (in comparison to the top figure). Why the variations for the higher flow rate are higher in the setup without the punched plate. For the black curves the error bars of the two flow rates seem to be comparable, whereas for the red curves, the error bars are much larger. Is there a reason for that?

9) In general: How many runs have been performed for the results and the averaging? How many test have been performed in total?

10) In line 358 a range for the particle sizes is given. In table 3, the mode value for NaCl-1 is given as 7.9 nm resulting in a range from 5.27 nm to 15.8 nm. In line 364, it is stated that the smallest measured aerosol size was 3 nm. Am I misunderstanding this or why the range can be extended in this case quite a lot?

11) Since the suggested methodology is intended to be an improvement to the previous techniques, I would have expected that there is a comparison to the previous techniques. If I saw it correctly, such a comparison is not shown in the manuscript.

Reviewer 2 Report

I have read with interest the article,,Development of a method to determine the fractional deposition efficiency of full scale HVAC and HEPA filter cassettes for nanoparticles ≥3.5 nm”. The article provides useful information regarding fractional deposition efficiency of full-scale HVAC and HEPA filters. The result can be useful in the context of the air decontamination, mostly nowadays when we interfere with the Sars-Cov2 threat. I recommend the publication of the article, after the correction of some references that have an error " Error! Reference source not found ". 
